# *Bacillus subtilis* and *Bifidobacteria bifidum* Fermentation Effects on Various Active Ingredient Contents in *Cornus officinalis* Fruit

**DOI:** 10.3390/molecules28031032

**Published:** 2023-01-19

**Authors:** Xiuren Zhou, Yimin Zhao, Lei Dai, Guifang Xu

**Affiliations:** 1Department of Biotechnology, School of Life Science and Technology, Henan Institute of Science and Technology, Hualan Road 90#, Xinxiang 453002, China; 2Guangxi Botanical Garden of Medicinal Plants, Changgang Road 189#, Nanning 530010, China

**Keywords:** fermentation, *Bifidobacteria*, *Bacillus*, active ingredient, *Cornus officinalis* fruit (COF)

## Abstract

Microbial fermentation has been widely used to improve the quality and functional composition of food and edibles; however, the approach has rarely been applied to traditional Chinese medicines. In this study, to understand the effect of microbial fermentation on the active ingredients of traditional Chinese medicines, we used *Bifidobacterium bifidum* and *Bacillus subtilis* to ferment the traditional Chinese medicine, *Cornus officinalis* fruit (COF), and determined the levels of active ingredients using HPLC (high-performance liquid chromatography). According to the results, both *B. subtilis* and *B. bifidum* substantially increased the amount of gallic acid in the COF culture broth after fermentation; however, the two species of bacteria had no effect on the loganin content. Moreover, the *B. subtilis* fermentation reduced the contents of ursolic acid and oleanolic acid in the COF broth, whereas the *B. bifidum* fermentation did not. This study contributes to a better understanding of the mechanism by which microbial fermentation alters the active ingredient levels of traditional Chinese medicines, and suggests that fermentation may potentially improve their functional ingredients.

## 1. Introduction

Microbial fermentation can increase or decrease certain ingredient contents in foods, fodders, and traditional medicines. *Lactobacillus*, which is one of the most studied microorganisms, produces various bioactive peptides with antioxidant, opioid antagonist, antiallergic, and hypotensive effects during the fermentation process [1,2]. The consumption of microbial-fermented foods also reduces the symptoms of lactose malabsorption and helps to eliminate *Helicobacter pylori* [2]. Microbial fermentation can reduce the antinutrients in pig feed and improve the feed absorption and intestinal flora [3]. In fermented food or feed, the vitamin A and carotenoids content, and tyrosinase inhibitory and antioxidant activities, are noticeably increased [4,5]. Consequently, fermentation is a promising method to ameliorate the quality ingredients of foods, fodders, and traditional herbs. In addition, microorganisms in human or other mammalian guts have different enzyme profiles that are involved in the metabolic pathways of carbohydrates, proteins, plant polyphenols, and vitamins, enhancing the efficiency of the absorption and utilization of these food components [6,7]. Microorganisms in the gastrointestinal tract were able to enhance the total release of phenolic compounds and flavonoids from Moringa leaves, thereby increasing the antioxidant activity when they were digested [8]. Fermented feeds can raise the productivity of pigs by improving their intestinal microflora to increase the utilization of the feed [9]. In vitro studies on the effects of the probiotic fermentation of food and traditional medicine ingredients may elucidate these changes in those components and ingredients within animal and human digestive systems.

*Cornus officinalis* fruit (COF), which is also called “Shan Zhu Yu”, is a commonly used Chinese herbal medicine for certain kidney and liver diseases [10]. COF is rich in active ingredients, such as ursolic acid, oleanolic acid, and loganin. These active compounds have different therapeutic effects, including antibacterial [11,12], anti-inflammatory [13,14], and anticancer [15,16,17] activities. Thus, increasing the active ingredient contents can improve the therapeutic effectiveness of herbal medicines. Su and Wang found that, with the increase in a wine yeast addition, the degradation rate of the polysaccharides in a COF fermentation broth was accelerated, and the tannin content was increased; however, there was no substantial effect on the loganin content [18]. Xu et al. report that an increase in yeast inoculation accelerated the polysaccharide degradation rate in COF, and that the fermentation reduced the dissolution resistance of ursolic acid but did not affect its level [19]. Park and Lee [20] demonstrated that a fermented COF extract promoted hair growth better than unfermented COF. However, we do not know what happens to the active compositions of COF during the fermentation process. Therefore, comparing the active ingredient contents of fermented COF with those of unfermented COF is a crucial step toward in elucidating the mechanism that underlies the improvement in the therapeutic effects of fermented COF via fermentation, as well as what happens to COF after it is fermented by the gut microbiota inside human or animal bodies.

Ursolic acid and oleanolic acid are effective at protecting against chemically induced liver damage, inflammation, tumor suppression, and the induction of apoptosis [21,22]. Loganin improves diabetic nephropathy [23] and drug-induced memory impairment [24], and it inhibits ex vivo inflammatory responses [25] and hydrogen peroxide-induced apoptosis [26]. Gallic acid has preventive and therapeutic activities in gastrointestinal, neuropsychological, obesity-related, metabolic, and cardiovascular diseases [27,28]. Ursolic acid, oleanolic acid, loganin, and gallic acid are the main bioactive components of *Cornus officinalis* fruit [29,30,31,32]. In this study, we fermented COF with *B. subtilis* and *B. bifidum*, and we determined the loganin, ursolic acid, oleanolic acid, and gallic acid contents of the COF culture broth at various fermentation times using high-performance liquid chromatography (HPLC). We demonstrated the microbial fermentation effects on the active ingredients of *Cornus officinalis*, providing a better understanding of the mechanism by which microbial fermentation changes the bioactive component contents of traditional Chinese herbs.

## 2. Results

### 2.1. Effects of B. subtilis and B. bifidum Fermentation on Ursolic Acid and Oleanolic Acid Contents

After the fermentation process, *B. bifidum* and *B. subtilis* had obviously different effects on the ursolic and oleanolic acid contents in the COF culture broth. Compared with those of unfermented COF, the ursolic and oleanolic acid contents of the COF culture broth fermented with *B. subtilis* were significantly (*p* < 0.05) decreased at different COF concentrations (Figure 1 and Figure 2) (Appendix A), which suggests that the bacteria can break down these compounds. Generally, the amounts of both active ingredients most rapidly decreased during the 12–24 h fermentation period; however, the contents exhibited no obvious changes during the 30–42 h fermentation period (Figure 1 and Figure 2), which indicates that the breakdown of the ursolic and oleanolic acid by *B. subtilis* was active in the early stage of fermentation, but became inactive in the later stage. In contrast, the *B. bifidum* fermentation had almost no effect on the ursolic and oleanolic acid contents in the COF culture broth. During the 6–42 h fermentation period, the contents of the two active compounds had no obvious alterations (Figure 1 and Figure 2), which suggests that the *B. bifidum* fermentation did not degrade the ursolic and oleanolic acid.

### 2.2. Effects of B. subtilis and B. bifidum Fermentation on the Loganin Content

The *B. subtilis* and *B. bifidum* fermentation did not have any obvious effects on the loganin amount in the COF culture broth. As listed in Figure 3 and Appendix A, although the loganin content showed some alteration in the *B. subtilis* culture broth with the different COF concentrations, the variation was not obvious, which implies that neither bacteria decomposed the loganin during the fermentation process.

### 2.3. Effects of B. subtilis and B. bifidum Fermentation on Gallic Acid Content

During the fermentation, both *B*. *subtilis* and *B*. *bifidum* were able to increase the gallic acid content in the COF culture broth. In Figure 4 and Appendix A, we can see that the gallic acid content in the COF culture broth fermented with *B. subtilis* was significantly (*p* < 0.05) promoted during the 6–18 h fermentation period at various COF concentration levels; however, there were no significant changes in the content during the 24–42 h fermentation period. Similarly, a significant (*p* < 0.05) increase in the gallic acid content in the COF culture broth fermented with *B. bifidum* generally occurred during the 6–18 h fermentation period; however, the content had no obvious alterations during the other fermentation periods (Figure 4), which suggests that the increases in the gallic acid caused by the two bacteria were active in the early stages of the fermentation, but became inactive in the later stages. The increase in the gallic acid level implied that it may be achieved by the degradation of certain compounds via *B. subtilis* or *B. bifidum* fermentation.

## 3. Discussion

In this study, both *B. subtilis* and *B. bifidum* increased the gallic acid level in the COF culture broth (Figure 4) (Table 1). Researchers have demonstrated that wine yeast fermentation increases the tannin content in COF culture broth [18], and that fermentation can increase the bioactive components of traditional herbal medicines. The antioxidant activities of traditional herbs can be substantially enhanced via microbial fermentation [5,8,33]. Wen et al. demonstrated that *Aspergillus oryzae* NCH 42 can improve the functional components, such as the total phenol content and antioxidant activity and content, of the traditional Chinese medicines *Trichosanthes kirilowii* Maxim, *Salvia miltiorrhiza* Bge, *Magnolia officinalis,* and *Glycyrrhizae radix* [34]. Green tea and *Houttuynia cordata* leaves fermented with *Lactobacillus paracasei,* subsp. *paracasei* NTU 101, contained higher levels of epigallocatechin gallate, epigallocatechin, and chlorogenic acid than with no fermentation [35]. According to the results, microbial fermentation can promote the bioactive component contents of traditional herbs.

In our study, we demonstrated that the *B. subtilis* fermentation reduced the ursolic and oleanolic acid levels in the COF culture broth (Figure 1 and Figure 2) (Table 1). Yeast fermentation can degrade the polysaccharides in COF fermentation broth [18,19]. Under optimized conditions, the tannin contents of traditional herbal Xaun Mugua fruits fermented with lactic acid bacteria were reduced by 78% compared with those of unfermented Xaun Mugua fruits [36]. The contents of hexanoic acid, octanoic acid, and butanoic acid were substantially decreased in noni juice fermented with *Acetobacter* sp. [37]. Through fermentation, the levels of antinutrients, toxic substances, and digestive enzyme inhibitors, in food are substantially reduced [38,39]. When traditional herbal medicines are fermented with microorganisms, the toxic components in them are decomposed, and their side effects are alleviated [40,41,42]. Fermentation reduces the levels of some components in food and traditional herbs.

The fermentation of *Acetobacter* sp. had no significant effect on most of the active ingredients in noni juice, although it substantially reduced the hexanoic, octanoic, and butanoic acid contents in the fruit [37]. In a traditionally fermented Chinese medicine, Massa Medicata Fermentata (MMF), the caffeic acid content did not change compared with that of unfermented MMF [43]. Yeast fermentation could not change the brucine and ursolic acid contents in COF [18,19]. In the current study, the *B. bifidum* fermentation did not alter the ursolic acid, oleanolic acid and loganin levels in the COF (Figure 1, Figure 2 and Figure 3) (Table 1), and the *B. subtilis* fermentation did not change the loganin levels in this fruit (Figure 3) (Table 1). Therefore, during microbial fermentation, not all of the fermented herbal component contents are affected.

Gallic acid is a potential precursor molecule for drug development [44], and its alkyl ester derivatives have anticancer and antioxidant abilities, neuroprotective functions, and induce the apoptosis of cancer cells [45,46,47]. In our study, both *B. subtilis* and *B. bifidum* could substantially increase the gallic acid levels of the COF culture broth. Therefore, before gallic acid is extracted from *Cornus officinalis* fruits as the lead molecule for drug development, the fermentation of these fruits using *B. subtilis* and *B. bifidum* may be one method of improving the gallic acid yield. Gallic acid also has potential preventive and therapeutic effects in oxidative-stress-related diseases, such as neurodegenerative disorders, cancer, cardiovascular diseases, and aging [48,49]. When COF is used as a functional food to prevent these diseases, *B. subtilis-* and *B. bifidum*-fermented COF have better effects than unfermented COF.

The initial decomposition stages of cyclic and aromatic compounds by bacteria and fungi require molecular oxygen for the oxidation of the compounds; therefore, aerobic conditions contribute to their breakdown [50,51,52]. In the current study, the pentacyclic triterpenoids ursolic acid and oleanolic acid from COF could be decomposed by *B. subtilis* under aerobic conditions; however, they could not be degraded by *B. bifidum* under anaerobic conditions, which suggests that oxygen may play a key role in the breakdown, by *B. subtilis,* of ursolic and oleanolic acids. Researchers have reported the involvement of different microorganisms in the degradation of the tannins in plant material to gallic acid during fermentation [53,54,55,56]. COF contains about 0.75% total tannins [57]. In this study, both *B. subtilis* and *B. bifidum* promoted the gallic acid levels in the COF culture broth after fermentation. The increased amount of gallic acid during fermentation may have been due to the degradation of tannin by these two bacteria.

In the present study, we investigated the effects of aerobic fermentation with *B. subtilis* and anaerobic fermentation by *B. bifidum* on the four active ingredients levels of COF, and we explored the mechanisms by which the bacterial fermentations influenced the active ingredients of the herbal medicine. The findings of this research contribute to a better understanding of the mechanisms by which microbial fermentation improves the active ingredients of traditional Chinese medicines. However, considerable work remains. We require investigations into the effects that these two bacteria have on the other components of COF, as well as on the active ingredients of other traditional Chinese medicines. In addition, we require studies on the molecular mechanisms by which these bacteria improve the active ingredients and disease-preventive effects of traditional herbal medicines.

## 4. Materials and Methods

### 4.1. Material Collection and Pretreatment

The material collection and pretreatment were performed according to the *Pharmacopoeia of the People’s Republic of China* (2015) and Song et al. (2018) [32,58]. In July 2020, fresh COF was harvested from a *Cornus officinalis* cultivation base of the Zhongjing Wanxi Pharmaceutical Co., Ltd. (Nanyang, China) and the kernels of the fruit were removed. The resultant flesh was mixed with rice wine at a ratio of 3:1 by mass, and kneaded. After rubbing, the flesh was steamed for 3 h, and then dried at 60 °C in an oven for 72 h. The dried flesh was ground into <3 mm particles. The powder was then prepared for the subsequent fermentation and extraction.

### 4.2. Reagents

The ursolic acid and loganin standards were purchased from Sigma-Aldrich Co. (St. Louis, MO, USA). Both the oleanolic acid and gallic acid standards were obtained from Sangon Biotech Co., Ltd. (Shanghai, China). The HPLC-grade methanol, acetonitrile, formic acid, and water were purchased from Thermo Fisher Scientific Inc. (Waltham, MA, USA). The medicines and consumables for the fermentation were purchased from the Shanghai Yuanye Biotech Co., Ltd. (Shanghai, China). The *B. subtilis* and *B. bifidum* strains were purchased from Shanghai Macklin Biochemical Co., Ltd. (Shanghai, China). Except for the liquid chromatography reagents, all of the other experimental chemicals were of analytical grade, and double-distilled water was applied.

### 4.3. COF Fermentation

The COF fermentation with *B. bifidum* was modified with reference to [59]. The COF fermentation was slightly modified with *B. subtilis* according to [60].

The COF was fermented not only with *B. bifidum*, which is an anaerobic bacterium, under anaerobic conditions, but also with *B. subtilis*, which is an aerobic bacterium, under aerobic conditions. Then, 180 mL of liquid media of the *B. bifidum* or *B. subtilis* was added to 500 mL flasks, and then mixed with 0, 1.2, 2.4, or 4.8 g of dried COF powder. In addition to water, the components of the *B. bifidum* medium were as follows: peptone: 10.0 g/L; beef extract powder: 8.0 g/L; yeast extract: 4.0 g/L; glucose: 20.0 g/L; sorbitan monooleate: 1 mL/L; dipotassium hydrogen phosphate: 2.0 g/L; sodium acetate: 5.0 g/L; triammonium citrate: 2.0 g/L; manganese sulfate heptahydrate: 0.2 g/L. One liter of the *B. subtilis* medium contained the following: water: 1000 mL; glucose: 20 g; peptone: 15 g; sodium chloride: 5 g; beef extract: 0.5 g. The pH values of the media mixed with COF powder were adjusted to 7.0 by adding sodium hydroxide or hydrochloric acid solution, which made up to 200 mL with the corresponding liquid medium. Afterwards, the flasks that contained the media were sterilized in an autoclave at 121 °C for 30 min. In a sterile environment, each flask containing the *B. bifidum* medium was incubated with 3 mL of 1 × 10^7^ cfu/mL *B. bifidum* culture, and all the flasks containing the *B. subtilis* medium were incubated with 2 mL of 2 × 10^7^ cfu/mL *B. subtilis* culture. The COF was fermented with *B. bifidum* at 38 °C in an anaerobic incubator (Defendor AMW1000; HUA YUE Enterprise holdings LTD, Guangzhou, China) for different durations: 0, 6, 12, 18, 24, 30, 36, or 42 h. The COF with *B. subtilis* was fermented at a speed of 120 rpm in a thermostatic shaker (HY-150; Wuhan Huicheng Biological Technology Co., Ltd., Wuhan, China) at 37 °C for different durations: 0, 6, 12, 18, 24, 30, 36, or 42 h. The samples without any COF were used as negative controls. The fermentation, with samples in triplicate, was then performed. When the specific fermentation times were reached, the fermentation processes of the samples were ended in a 0–4 °C environment. Then, the culture broths were centrifuged at 7500 rpm at 4 °C for 30 min, and 120 mL of the supernatant was aspirated, aliquoted into 40 mL portions, and placed in a 4 °C refrigerator for the subsequent assays.

### 4.4. Extraction and Determination of Ursolic Acid and Oleanolic Acid

Next, 40 mL of the resultant fermentation broth supernatant was evaporated under reduced pressure in a rotary evaporator, and concentrated to form a paste residue. The residue was extracted twice with 50 mL of petroleum ether. Next, the residue was added to 40 mL of 95% alcohol, extracted using ultrasound at 60 °C for 90 min, and then concentrated into a paste residue under a rotary evaporator. The resulting residue was dissolved in water, and the solution was extracted three times with chloroform. Afterwards, 20 mL of supernatant was obtained, filtered through a 0.45 μm filter, and evaporated in a rotary vacuum evaporator. The resultant residue was dissolved in 20 mL of methanol. A 20 μL sample was injected into an HPLC system (LC-2010HT; Shimadzu Corporation, Kyoto, Japan). The ursolic acid and oleanolic acid assays were performed at 218 nm using an ODS-2 C18 analytical column (Thermo Scientific, MA, USA) and a mobile phase comprising methanol, water, glacial acetic acid, and triethylamine at a ratio of 90:9.9:0.06:0.04 (*v*/*v*). The flow rate was 0.5 mL/min, and the column temperature was 30 °C. The ursolic acid and oleanolic acid of the samples were identified by comparing the retention times of the peaks in the samples with those of standard ursolic acid and oleanolic acid (Appendix A). The ursolic acid and oleanolic acid contents in the test samples were estimated by measuring and analyzing the peak areas of the samples based on the ursolic acid and oleanolic acid standards. The unfermented COF and control samples were treated in the same manner. The determination of the ursolic acid and oleanolic acid was conducted according to Tian et al. [61].

### 4.5. Extraction and Determination of Loganin

Next, 40 mL of the culture broth supernatant was evaporated in a rotary evaporator. The resulting residue was extracted twice with 50 mL of petroleum ether, and then dissolved in 40 mL of 80% methanol. The solution was heated under reflux for 1 h and then left to cool. Afterwards, the amount of solution lost was supplemented during the reflux with 80% methanol, and filtered through a 0.45 μm filter. The resulting solution was used as the sample to be tested. A 20 μL sample was used for the high-performance liquid chromatography (LC-2010HT; Shimadzu Corporation, Kyoto, Japan), which was run at 240 nm using an ODS2C18 analytical column and a mobile phase formed of methanol and water at a ratio of 75:25 (*v*/*v*). The column temperature was 35 °C. The loganin in the samples was identified by comparing the retention times of the peaks in the samples with those of standard loganin. The loganin content in the test samples was estimated by measuring and analyzing the peak areas of the sample and standard loganin (Appendix A). The loganin contents in the unfermented COF and control samples were determined in the same manner as described in [32].

### 4.6. Extraction and Determination of Gallic Acid

Next, 40 mL of the broth supernatant was evaporated using a rotary evaporator, and then the paste residue was washed with 50 mL of petroleum ether to remove the lipid compounds. The resultant residue was dissolved in 40 mL of methanol, and the solution was filtered through a 0.45 μm filter. The solution was used as an assayed sample. A 20 μL sample was injected into the HPLC system (LC-2010HT; Shimadzu Corporation, Kyoto, Japan). The determination of the gallic acid in the test samples was performed at 30 °C using an ODS-2 C18 analytical column, with a mobile phase comprising a methanol and phosphoric acid solution (pH = 3) at a ratio of 95:5 (*v*/*v*). The column temperature was set at 30 °C, and the flow rate was 0.5 mL per minute. Gallic acid at 272 nm was detected. The gallic acid in the culture broth was identified by comparing the retention times of the peaks in the samples with those of standard gallic acid (Appendix A). The quantitation of the gallic acid samples was performed by measuring and analyzing the peak areas of the sample and standard gallic acid. The identification and quantitation of the gallic acid of the unfermented COF and control samples were performed in the same manner. The gallic acid content was determined according to the method described in [62].

### 4.7. Statistical Analysis

All the data for normality and constant variance were tested prior to the statistical analysis. One-way analysis of variance (ANOVA) and least significant difference (LSD) multiple comparison tests were used to detect the differences between the groups of active compound contents. The resulting values were expressed as means ± standard errors. The mean differences were considered to be statistically significant at *p* < 0.05. The statistical analysis was performed using SPSS 17.0 for Windows (SPSS Inc., Chicago, IL, USA).

## 5. Conclusions

Microbial fermentation improves not only the nutritional compositions of food and feedstuff, but also the bioactive compositions of traditional herbal medicines. Aerobic *B. subtilis* fermentation and anaerobic *B. bifidum* fermentation increased the gallic acid levels in COF; however, they had no effect on the loganin content. *B. subtilis* reduced the levels of ursolic acid and oleanolic acid; however, *B. bifidum* did not. In addition, the degradation of the ursolic and oleanolic acids by *B. subtilis* may have been related to the aerobic conditions, and the increases in gallic acid by *B. subtilis* and *B. bifidum* may have been the result of tannin degradation. These findings contribute to a better understanding of the mechanisms by which microbial fermentation improves the bioactive components of traditional herbal medicines, and they also imply that microbial fermentation is a potential approach to improving the active ingredients and disease prevention effects of traditional herbal medicines.

## Figures and Tables

**Figure 1 molecules-28-01032-f001:**
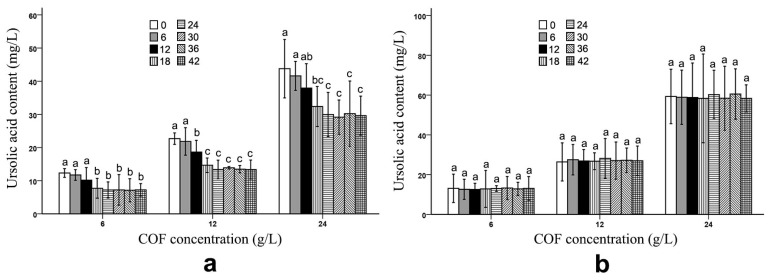
Ursolic acid content in *Cornus officinalis* fruit (COF) culture broth at different concentrations and fermentation times of *Bacillus subtilis* and *Bifidobacterium bifidum*. (**a**) Ursolic acid content in COF culture broth at different concentrations and fermentation duration of *B. subtilis*. (**b**) Ursolic acid content in COF culture broth at different concentrations and fermentation duration of *B. bifidum*. Data are the means ± SD from three independent tests. The different letters (a, ab, b, bc, and c) on the error bars indicate significant differences at *p* < 0.05 between the ursolic acid contents of different fermentation durations on the same COF concentration and strain. The significant differences (*p* < 0.05) of the mean values were detected by T-test.

**Figure 2 molecules-28-01032-f002:**
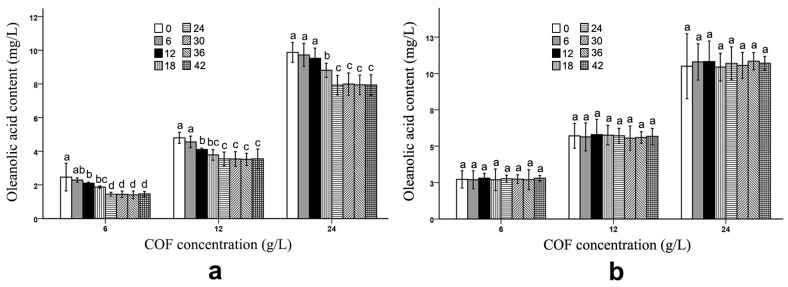
Oleanolic acid content in *Cornus officinalis* fruit (COF) culture broth at different concentrations and fermentation times of *Bacillus subtilis* and *Bifidobacterium bifidum*. (**a**) Oleanolic acid content in COF culture broth at different concentrations and fermentation duration of *B. subtilis*. (**b**) Oleanolic acid content in COF culture broth at different concentrations and fermentation duration of *B. bifidum*. Data are the means ± SD from three independent tests. Different letters (a, ab, b, bc, c, and d) on the error bars indicate significant differences at *p* < 0.05 between the oleanolic acid contents of different fermentation durations on the same COF concentration and strain. The significant differences (*p* < 0.05) of the mean values were detected by T-test.

**Figure 3 molecules-28-01032-f003:**
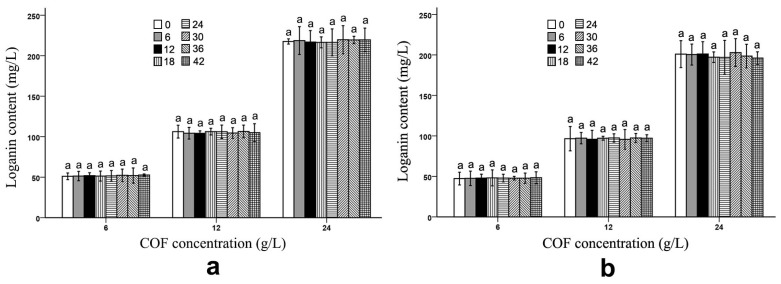
Loganin content in *Cornus officinalis* fruit (COF) culture broth at different concentrations and fermentation times of *Bacillus subtilis* and *Bifidobacterium bifidum*. (**a**) Loganin content in COF culture broth at different concentrations and fermentation duration of *B. subtilis*. (**b**) Loganin content in COF culture broth at different concentrations and fermentation duration of *B. bifidum*. Data are the means ± SD from three independent tests. Letters (a) on the error bars indicate significant differences at *p* < 0.05 between the loganin contents of different fermentation durations on the same COF concentration and strain. The significant differences (*p* < 0.05) of the mean values were detected by T-test.

**Figure 4 molecules-28-01032-f004:**
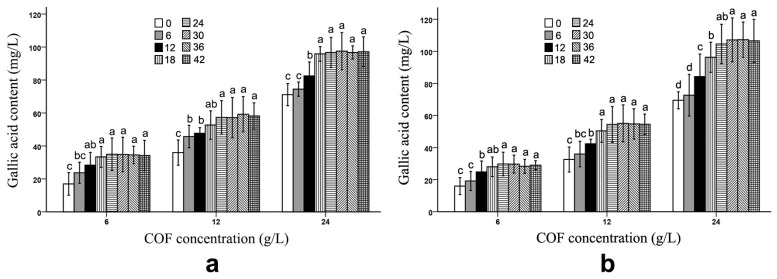
Gallic acid content in *Cornus officinalis* fruit (COF) culture broth at different concentrations and fermentation times of *Bacillus subtilis* and *Bifidobacterium bifidum*. (**a**) Gallic acid content in COF culture broth at different concentrations and fermentation duration of *B. subtilis*. (**b**) Gallic acid content in COF culture broth at different concentrations and fermentation duration of *B. bifidum*. Data are the means ± SD from three independent tests. Different letters (a, ab, b, bc, c, and d) on the error bars indicate significant differences at *p* < 0.05 between the gallic acid contents of different fermentation durations on the same COF concentration and strain. The significant differences (*p* < 0.05) of the mean values were detected by T-test.

**Table 1 molecules-28-01032-t001:** Alteration of four active ingredients in COF after fermentation with *B. subtilis* and *B. bifidum*.

Bacterial Strain	Active Ingredients of COF			
	Ursolic Acid	Oleanolic Acid	Loganin	Gallic Acid
*B. subtilis*	Fall	Fall	No change	Rise
*B. bifidum*	No change	No change	No change	Rise

Rise, Fall or No Change indicate the increase, decrease or no change in the content of an active ingredient of COF after fermentation with *B. subtilis* and *B. bifidum*.

## Data Availability

The data presented in this study are available upon request from the corresponding author.

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
