# Peer review of "Bacillus subtilis and Bifidobacteria bifidum Fermentation Effects on Various Active Ingredient Contents in Cornus officinalis Fruit"

_molecules, 2023, doi:10.3390/molecules28031032_

Round 1

Reviewer 1 Report

The present manuscript needs major revision before been accepted for publication. I do think the English language needs to be improved throughout the manuscript as well as several terminologies. Please see my detailed comments below:

 There is no aim or objective, please include one.

 Lines 11-12Please rewrite all this part, particularly the first sentence.

 Lines 18-21Please rewrite all this part, expression unclear.

 Lines 27-28How fermentation improves nutritional value and efficacy?

 Lines 30-31Please rewrite, not clear

 Lines 62-63Please provide the specific extracellular enzyme decomposition, because your statement is invalid.

 Lines 66-67It is not possible to state these unless you have data that validates this statement. Please rewrite.

 Lines 94-95Please rewrite, not clear. B. bifidum increased the content of that active ingredient.

 Lines 102-103Please provide the measured values of the two bacterial growth stages. Your statement is invalid because your results have not been verified by the actual target measurements of the growth stage.

 Lines 113-114I don 't think there is much research, please rewrite.

 Lines 111-124Please discuss it again by comparing the research results of others.

 Lines 125-126which results suggest that choosing the right fermentation microorganism is a crucial step during the improvement of active compounds or curing efficacy of COF by fermentation?

 Lines 127-130:I do not think that the degradation of ursolic acid and oleanolic acid by Bacillus subtilis can be concluded that Bacillus subtilis is not suitable for increasing the content of active compounds and weakening the curing effect of COF. Please rewrite.

 Lines 135-136:Please specify.

 Lines 137-138:Please add references supporting your results.

 Lines 141-144:Please add references supporting your results.

 Lines 145-146:I don 't think so. Please add references supporting your results.

 Lines 147-150:This can 't even be speculated, because it 's much more complicated to cause a reduction or increase in the efficacy of traditional drugs. Please rewrite.

 Lines 155-156:Please add references supporting your methods.

 Line 172: Correct ‘500-mL’ in ‘500 mL’.

Line 176: Correct ‘manganese sulphate 7H2O’ in ‘Manganese sulfate heptahydrate’.

 Line 182: Correct ‘1 × 107’ in ‘1 × 107’.

 Line 183: Correct ‘2 × 107’ in ‘2 × 107’.

 Line 202: Correct ‘0.45-μm’ in ‘0.45 μm’.

 Line 204: Correct ‘20-μL’ in ’20 μL’.

 Line 219: Correct ‘0.45-μm’ in ‘0.45 μm’.

 Line 232: Correct ‘0.45-μm’ in ‘0.45 μm’.

 Lines 251-262:Please rewrite all these contents. The conclusion is a high-level overview of the experimental results, not a rewriting of the results.

Author Response

Dear Reviewer,

We appreciate the time and effort that you have dedicated to providing valuable feedback on our manuscript. We are grateful to you for your insightful comments on our paper. We have been able to incorporate changes to reflect the suggestions provided by your. We have highlighted the changes within the manuscript with track changes. The English language have been improved throughout the manuscript by MDPI English Editing according to your suggestion.

Here is a point-by-point response to your comments and concerns.

Comment 1: There is no aim or objective, please include one.

Response 1: Agree. We have added the aim.

Comment 2: Lines 11-12:Please rewrite all this part, particularly the first sentence.

Response 2: Thank you for pointing this out. We agree with this comment and rewrite this part.

Comment 3: Lines 18-21:Please rewrite

Response 3: We agree with this and have re

Comment 4: all this part, expression unclear.

Response 4: Agree. We have rewritten the part and tried to express them clearly.

Comment 5: Lines 27-28:How fermentation improves nutritional value and efficacy?

Response 5: Agree. We have specifically described how fermentation improves nutritional value and efficacy.

Comment 6: Lines 30-31:Please rewrite, not clear

Response 6: Agree. We have rewritten this part and tried to express them clearly.

Comment 7: Lines 62-63:Please provide the specific extracellular enzyme decomposition, because your statement is invalid.

Response 7: Thank you for pointing this out. We have deleted this part and modified the description.

Comment 8: Lines 66-67:It is not possible to state these unless you have data that validates this statement. Please rewrite.

Response 8: Agree. We have deleted this statement and rewritten them.

Comment 9: Lines 94-95:Please rewrite, not clear. B. bifidum increased the content of that active ingredient.

Response 9: Agree. We have rewritten this part and tried to express them clearly.

Comment 10: Lines 102-103:Please provide the measured values of the two bacterial growth stages. Your statement is invalid because your results have not been verified by the actual target measurements of the growth stage.

Response 10: Thank you for pointing this out. We have rewritten this part because we did not carry out the actual target measurements of the growth stage.

Comment 11: Lines 113-114:I don 't think there is much research, please rewrite.

Response 11: Agree. We have rewritten this part and clearly expressed them.

Comment 12: Lines 111-124:Please discuss it again by comparing the research results of others.

Response 12: Agree. We have discussed this part again by comparing the research results of other more literatures.

Comment 13: Lines 125-126:which results suggest that choosing the right fermentation microorganism is a crucial step during the improvement of active compounds or curing efficacy of COF by fermentation?

Response 13: Agree. We have deleted this part and rewritten them.

Comment 14: Lines 127-130:I do not think that the degradation of ursolic acid and oleanolic acid by Bacillus subtilis can be concluded that Bacillus subtilis is not suitable for increasing the content of active compounds and weakening the curing effect of COF. Please rewrite.

Response 14: Agree. We have deleted this part and rewritten them.

Comment 15: Lines 135-136:Please specify.

Response 15: Agree. We have rewritten this part.

Comment 16: Lines 137-138:Please add references supporting your results.

Response 16: Agree. We have rewritten this part and added references supporting the results.

Comment 17: Lines 141-144:Please add references supporting your results.

Response 17: Agree. We have rewritten this part and added references supporting the results.

Comment 18: Lines 145-146:I don 't think so. Please add references supporting your results.

Response 18: Agree. We have changed our view and rewritten this part.

Comment 19: Lines 147-150:This can 't even be speculated, because it 's much more complicated to cause a reduction or increase in the efficacy of traditional drugs. Please rewrite.

Response 19: Agree. We have changed our view and rewritten this part.

Comment 20: Lines 155-156:Please add references supporting your methods.

Response 20: Agree. We have added references supporting the our method.

Comment 21: Line 172: Correct ‘500-mL’ in ‘500 mL’.

Response 21: Agree. We have done.

Comment 22: Line 176: Correct ‘manganese sulphate 7H2O’ in ‘Manganese sulfate heptahydrate’.

Response 22: Agree. We have done.

Comment 23: Line 182: Correct ‘1 × 107’ in ‘1 × 107’.

Response 23: Agree. We have done.

Comment 24: Line 183: Correct ‘2 × 107’ in ‘2 × 107’.

Response 24: Agree. We have done.

Comment 25: Line 202: Correct ‘0.45-μm’ in ‘0.45 μm’.

Response 25: Agree. We have done.

Comment 26:Line 204: Correct ‘20-μL’ in ’20 μL’.

Response 26: Agree. We have done.

Comment 27: Line 219: Correct ‘0.45-μm’ in ‘0.45 μm’.

Response 27: Agree. We have done.

Comment 28: Line 232: Correct ‘0.45-μm’ in ‘0.45 μm’.

Response 28: Agree. We have done.

Comment 29: Lines 251-262:Please rewrite all these contents. The conclusion is a high-level overview of the experimental results, not a rewriting of the results.

Response 29: Agree. We have rewritten this section and expressed them in a high-level overview of the results.

Sincerely,

Xiuren Zhou

Jan. 5, 2003

Reviewer 2 Report

Zhou and co-workers describe the effect of fermentation using two probiotic bacterial strains (Bacillus subtilis and Bifidobacterium bifidum) on a preparation made of Cornus officinalis fruits. The effect of fermentation on the concentration of selected bioactive compounds was assessed, namely ursolic acid, oleanolic acid and gallic acid, as well as loganin.

Major comments:

Since the authors are specifically focusing on these compounds, a more detailed description of their effects, which make their study relevant, should be provided either within the introduction or in the discussion of the results. I suggest the authors rewrite the manuscript to include this information.

Although the results seem promising and interesting, the amount of work and the conclusions are quite limited. The main goal seems to be solely to see if these two strains would have a different impact on the selected compounds, which would be expectable, considering that they are different species, that even present distinct metabolic behavior (aerobic vs anaerobic, for instance). As it is not clear why those specific compounds were selected, it seems that more bioactive compounds could have been evaluated to provide more insights on the effects of the fermentation effects. Alternatively, other parameters could be assessed upon fermentation to understand other changes due to this process, namely, microbiota modulation or bioactivity testing, to expand the relevance of the paper.

Minor comments:

The authors chose to present the results in a table format. A graphical representation would provide an easier format to compare the conditions, while the tables could still be presented as supplementary material for consultation.

A small table/image summarizing all the effects, considering both strains and the compounds analyzed would be helpful for the reader to understand the overall message of the paper.

Author Response

Dear Reviewer,

We appreciate the time and effort that you have dedicated to providing valuable feedback on our manuscript. We are grateful to you for your insightful comments on our paper. We have been able to incorporate changes to reflect the suggestions provided by your. We have highlighted the changes within the manuscript with track changes.

Here is a point-by-point response to your comments and concerns.

Comment 1: Since the authors are specifically focusing on these compounds, a more detailed description of their effects, which make their study relevant, should be provided either within the introduction or in the discussion of the results. I suggest the authors rewrite the manuscript to include this information.

Response 1:  Thank you for pointing this out. We have rewritten part of the introduction and all the discussion, and provided a more detailed description and discussion of the four active components and their effects.

Comment 2: Although the results seem promising and interesting, the amount of work and the conclusions are quite limited. The main goal seems to be solely to see if these two strains would have a different impact on the selected compounds, which would be expectable, considering that they are different species, that even present distinct metabolic behavior (aerobic vs anaerobic, for instance). As it is not clear why those specific compounds were selected, it seems that more bioactive compounds could have been evaluated to provide more insights on the effects of the fermentation effects. Alternatively, other parameters could be assessed upon fermentation to understand other changes due to this process, namely, microbiota modulation or bioactivity testing, to expand the relevance of the paper.

Response 2: Thank you for these suggestions. We chose these four active substances because they are the main active ingredients of the fruit of Cornus officinalis, and these active ingredients have been reported to have therapeutic or preventive effects on different diseases such as diabetic nephropathy, drug-induced memory impairment, anti-inflammatory, tumor suppression, neuropsychological, obesity, metabolic, etc. We have explained the reasons for selecting these active ingredients in the first half of the last paragraph of the introduction. It would have been interesting to evaluate other parameters of the fermentation process to understand other changes. However, in the case of our study, it seems slightly out of scope because we are mainly concerned about the influence of two kinds of bacteria with different fermentation conditions on the four active components of the fruit of Cornus officinalis. In addition, We have rewritten most of the introduction and discussion, and added literature to support our results.

Comment 3: The authors chose to present the results in a table format. A graphical representation would provide an easier format to compare the conditions, while the tables could still be presented as supplementary material for consultation.

Response 3: Agree. We have presented the results in a histogram format and provided the tables as supplementary materials.

Comment 4: A small table/image summarizing all the effects, considering both strains and the compounds analyzed would be helpful for the reader to understand the overall message of the paper.

Response 4: Agree. We have provided a table to help readers understand the overall message of the paper.

Sincerely,

Xiuren Zhou

Jan. 5, 2003

Reviewer 3 Report

General comments

The manuscript deals with the effects of Bacillus subtilis and Bifidobacteria bifidum fermentation on the concentrations of ursolic, oleanolic, gallic acids and loganin in Cornus officinalis fruit.

Although the work could be of interest, the following issues limit the scientific merit of the paper:

1. The results are not discussed in depth and some explanations are speculative.

2. In general, the authors only showed the results obtained but without giving the explanations of these results. In addition, a clear conclusion of the results obtained is not given.

3. On the other hand, the Materials and methods section should be carefully revised by the authors to correct some technical errors detected in this section.

Other considerations are as follows:

1. Introduction.

1.1. Lines 40–44: Some parts of the Introduction section seems to be a literature review rather than an overview of the state of the art of fermentation of Cornus officinalis fruit. For example:

What were the main results obtained by Su and Wang [19] during the fermentation condition and determination of loganin?

What were the main results obtained by Xu et al. [20] during the extraction of ursolic acid from this herb through fermentation?

2. Results.

2.1. Subsection 2.1. Effects of B. subtilis and B. bifidum Fermentation on Ursolic Acid and Oleanolic Acid Contents:

2.1.1. Lines 62–63: What extracellular enzymes may have produced by B. bifidum and B. subtilis to break the ursolic acid and oleanolic acids?

2.1.2. Lines 66–68: Why did the authors said that “the regular change in the ability of B. subtilis to affect the ursolic acid and oleanolic acid contents in COF couture broth was essentially consistent with the growth phases of the bacterium?. In fact, the authors did not show the growth curve for this bacterium.

2.1.3. Lines 69–71: Why during the 6–42 h 69 fermentation period didn´t the contents of the two active compounds show obvious alterations?

2.1.4. Tables 1 and 2:

- The kinetics data shown in Tables 1 and 2 are difficult to interpret. So that these data should be better presented in a graph for a better comprehension.

- The authors indicate in Tables 1 and 2 that “Different letters following each value indicate significant differences (P < 0.05) in mean values”. Please specify if the mean values compared are in columns or rows, and the test used to detect significant differences (P < 0.05) in mean values.

- Why the initial ursolic (Table 1) and oleanolic acid (Table 2) contents in Cornus officinalis fruit are different in the fermentations with Bacillus subtilis and Bifidobacterium bifidum?

In the case of ursolic acid (Table 1):

- for 6(Bs) and 6(Bb) the initial ursolic acid concentrations were 12.32±0.65 and 13.12±3.56 g/L.

- for 12(Bs) and 12(Bb) the initial ursolic acid concentrations were 22.69±0.87 and 26.40±4.79 g/L.

- for 24(Bs) and 24(Bb) the initial ursolic acid concentrations were 43.77±4.40 and 59.33±6.84 g/L.

In the case of oleanolic acid (Table 2):

- for 6(Bs) and 6(Bb) the initial ursolic acid concentrations were 2.46±0.41 and 2.72±0.30 g/L.

- for 12(Bs) and 12(Bb) the initial ursolic acid concentrations were 4.80±0.16 and 5.71±0.43 g/L.

- for 24(Bs) and 24(Bb) the initial ursolic acid concentrations were 9.87±0.30 and 10.50±1.12 g/L.

2.2. Subsections 2.2 and 2.3:

2.2.1. Lines 84–87: Why neither B. subtilis fermentation nor B. bifidum fermentation didn´t show an obvious effect on the amount of loganin in the COF culture broth?

2.2.2. Tables 3 and 4:

- The kinetics data shown in Tables 3 and 4 are difficult to interpret. So that these data should be better presented in a graph for a better comprehension.

- The authors indicate in Tables 3 and 4 that “Different letters following each value indicate significant differences (P < 0.05) in mean values”. Please specify if the mean values compared are in columns or rows, and the test used to detect significant differences (P < 0.05) in mean values.

- Why the initial loganin (Table 3) and gallic acid (Table 2) contents in Cornus officinalis fruit are different in the fermentations with Bacillus subtilis and Bifidobacterium bifidum?

In the case of loganin (Table 3):

- for 6(Bs) and 6(Bb) the initial loganin concentrations were 51.10±2.05 and 47.30 ±3.93 g/L.

- for 12(Bs) and 12(Bb) the initial loganin concentrations were 106.22 ±4.32 and 96.56±7.50 g/L.

- for 24(Bs) and 24(Bb) the initial loganin concentrations were 217.57 ±4.40 and 200.98±8.30 g/L.

In the case of oleanolic acid (Table 4):

- for 6(Bs) and 6(Bb) the initial gallic acid concentrations were 16.96±3.43 and 16.00±2.64 g/L.

- for 12(Bs) and 12(Bb) the initial gallic acid concentrations were 35.96±3.83 and 32.60±3.90 g/L.

- for 24(Bs) and 24(Bb) the initial gallic acid concentrations were 71.13±3.36 and 69.50±2.66 g/L.

2.2.3. Lines 94–103: Why did the content of gallic acid increase in the different fermentations with Bacillus subtilis and Bifidobacterium bifidum?

3. Discussion.

3.1. Lines 111–114: This information is repetitive because it was provided in the Introduction section (lines 25–30).

3.2. Lines 114–118: This information is repetitive because it was provided in the Results section.

3.3. Lines 120–123: Which active ingredients can be improved during the fermentation of herbal medicines?

3.4. Lines 127–134: This information is repetitive because it was provided in the Results section.

3.5. Lines 135–138: Which complex macromolecular compounds can be degraded into micromolecular ones during fermentation?

3.6. Lines 138–150: This is a repetition of the same statements indicated in the previous sentences.

Generally, in this section, the authors continuously repeat the results without clearly explaining the results obtained. In fact, this section is very boring and very difficult to interpret and understand. 

In fact, the authors did not explain the increase or decrease of the different active compounds that they studied during fermentations with Bacillus subtilis and Bifidobacterium bifidum.

4. Materials and Methods

4.1. Line 155: Why was the resultant flesh mixed with rice wine? Why was the resultant flesh mixed with rice wine at a ratio of 3:1 by mass?

4.2. Line 170–171: Why was the COF fermented by B. bifidum under anaerobic conditions? Why was the COF fermented by B. subtilis under aerobic conditions?

4.3. Lines 181–183: Why were different inoculum concentrations of B. bifidum (1 × 107 cfu/mL) and B. subtilis (2 × 107 cfu/mL) used to ferment the COF at different initial concentrations?

5. Conclusions

Lines 251–262: These sentences are repetitive. No clear conclusion has been expressed in this section.

Author Response

Dear Reviewer,

We appreciate the time and effort that you have dedicated to providing valuable feedback on our manuscript. We are grateful to you for your insightful comments on our paper. We have been able to incorporate changes to reflect the suggestions provided by your. We have highlighted the changes within the manuscript with track changes.

Here is a point-by-point response to your comments and concerns.

Comment 1: 1. The results are not discussed in depth and some explanations are speculative.

Response 1: Thank you for pointing this out. We have rewritten the discussion and tried to discussed in depth. We also added more references to support our results.

Comment 2: 2. In general, the authors only showed the results obtained but without giving the explanations of these results. In addition, a clear conclusion of the results obtained is not given.

Response 2: Thank you for pointing this out. We have rewritten the discussion and given more explanation of our results. We have rewritten the conclusion and tried to provide a clear conclusion.

Comment 3: 3. On the other hand, the Materials and methods section should be carefully revised by the authors to correct some technical errors detected in this section.

Response 3: Agree. We have corrected the errors.

Other considerations are as follows:

  1. Introduction.

Comment 4: 1.1. Lines 40–44: Some parts of the Introduction section seems to be a literature review rather than an overview of the state of the art of fermentation of Cornus officinalis fruit. For example:

What were the main results obtained by Su and Wang [19] during the fermentation condition and determination of loganin?

What were the main results obtained by Xu et al. [20] during the extraction of ursolic acid from this herb through fermentation?

Response 4: Agree. We have rewritten the parts in the Introduction and specifically described the main results of the cited literature.

  1. Results.

2.1. Subsection 2.1. Effects of B. subtilis and B. bifidum Fermentation on Ursolic Acid and Oleanolic Acid Contents:

Comment 5: 2.1.1. Lines 62–63: What extracellular enzymes may have produced by B. bifidum and B. subtilis to break the ursolic acid and oleanolic acids?

Response 5: Thank you for pointing this out. We have deleted this part and modified the description.

Comment 6: 2.1.2. Lines 66–68: Why did the authors said that “the regular change in the ability of B. subtilis to affect the ursolic acid and oleanolic acid contents in COF couture broth was essentially consistent with the growth phases of the bacterium?. In fact, the authors did not show the growth curve for this bacterium.

Response 6: Thank you for pointing this out. We have rewritten this part because we did not carry out the actual target measurements of the growth stage.

Comment 7: 2.1.3. Lines 69–71: Why during the 6–42 h 69 fermentation period didn´t the contents of the two active compounds show obvious alterations?

Response 7: Thank you for pointing this out. We have explained this.

2.1.4. Tables 1 and 2:

Comment 8: - The kinetics data shown in Tables 1 and 2 are difficult to interpret. So that these data should be better presented in a graph for a better comprehension.

Response 8: Agree. We have provided these results in a graph format instead of a table format.

Comment 9: - The authors indicate in Tables 1 and 2 that “Different letters following each value indicate significant differences (P < 0.05) in mean values”. Please specify if the mean values compared are in columns or rows, and the test used to detect significant differences (P < 0.05) in mean values.

Response 9: Agree. We have added these contents in the legend.

Comment 10: - Why the initial ursolic (Table 1) and oleanolic acid (Table 2) contents in Cornus officinalis fruit are different in the fermentations with Bacillus subtilis and Bifidobacterium bifidum?

In the case of ursolic acid (Table 1):

- for 6(Bs) and 6(Bb) the initial ursolic acid concentrations were 12.32±0.65 and 13.12±3.56 g/L.

- for 12(Bs) and 12(Bb) the initial ursolic acid concentrations were 22.69±0.87 and 26.40±4.79 g/L.

- for 24(Bs) and 24(Bb) the initial ursolic acid concentrations were 43.77±4.40 and 59.33±6.84 g/L.

In the case of oleanolic acid (Table 2):

- for 6(Bs) and 6(Bb) the initial ursolic acid concentrations were 2.46±0.41 and 2.72±0.30 g/L.

- for 12(Bs) and 12(Bb) the initial ursolic acid concentrations were 4.80±0.16 and 5.71±0.43 g/L.

- for 24(Bs) and 24(Bb) the initial ursolic acid concentrations were 9.87±0.30 and 10.50±1.12 g/L.

Response 10: Thank you for the suggestions. The difference of initial contents between Bacillus subtilis and Bifidobacterium bifidum fermentation is due to the fact that two different batches of Cornus officinalis fruits were used for Bacillus subtilis fermentation and Bifidobacterium bifidum fermentation, respectively. We have not mixed the two batches of fruits before the fermentation. We have believed that this difference in initial values had no effect on the analysis and interpretation of the results, so we did not make a note. We highly appreciate the reviewers' comments and have included a note explaining this in each table legend.

2.2. Subsections 2.2 and 2.3:

Comment 11: 2.2.1. Lines 84–87: Why neither B. subtilis fermentation nor B. bifidum fermentation didn´t show an obvious effect on the amount of loganin in the COF culture broth?

Response 11: Thank you for pointing this out. We have explained this.

2.2.2. Tables 3 and 4:

Comment 12: - The kinetics data shown in Tables 3 and 4 are difficult to interpret. So that these data should be better presented in a graph for a better comprehension.

Response 12: Agree. We have provided these results in a graph format instead of a table format.

Comment 13: - The authors indicate in Tables 3 and 4 that “Different letters following each value indicate significant differences (P < 0.05) in mean values”. Please specify if the mean values compared are in columns or rows, and the test used to detect significant differences (P < 0.05) in mean values.

Response 13: Agree. We have added these contents in the legend.

Comment 14: - Why the initial loganin (Table 3) and gallic acid (Table 2) contents in Cornus officinalis fruit are different in the fermentations with Bacillus subtilis and Bifidobacterium bifidum?

In the case of loganin (Table 3):

- for 6(Bs) and 6(Bb) the initial loganin concentrations were 51.10±2.05 and 47.30 ±3.93 g/L.

- for 12(Bs) and 12(Bb) the initial loganin concentrations were 106.22 ±4.32 and 96.56±7.50 g/L.

- for 24(Bs) and 24(Bb) the initial loganin concentrations were 217.57 ±4.40 and 200.98±8.30 g/L.

In the case of oleanolic acid (Table 4):

- for 6(Bs) and 6(Bb) the initial gallic acid concentrations were 16.96±3.43 and 16.00±2.64 g/L.

- for 12(Bs) and 12(Bb) the initial gallic acid concentrations were 35.96±3.83 and 32.60±3.90 g/L.

- for 24(Bs) and 24(Bb) the initial gallic acid concentrations were 71.13±3.36 and 69.50±2.66 g/L.

Response 14: Thank you for the suggestions. The difference of initial contents between Bacillus subtilis and Bifidobacterium bifidum fermentation is due to the fact that two different batches of Cornus officinalis fruits were used for Bacillus subtilis fermentation and Bifidobacterium bifidum fermentation, respectively. We have not mixed the two batches of fruits before the fermentation. We have believed that this difference in initial values had no effect on the analysis and interpretation of the results, so we did not make a note. We highly appreciate the reviewers' comments and have included a note explaining this in each table legend.

Comment 15: 2.2.3. Lines 94–103: Why did the content of gallic acid increase in the different fermentations with Bacillus subtilis and Bifidobacterium bifidum?

Response 15: Thank you for pointing this out. We have explained this.

  1. Discussion.

Comment 16: 3.1. Lines 111–114: This information is repetitive because it was provided in the Introduction section (lines 25–30).

Response 16: Thank you for the comment. We have rewritten this parts and added more references to support our results.

Comment 17: 3.2. Lines 114–118: This information is repetitive because it was provided in the Results section.

Response 17: Thank you for the comment. We have rewritten this parts and added more references to support our results.

Comment 18: 3.3. Lines 120–123: Which active ingredients can be improved during the fermentation of herbal medicines?

Response 18: We agree with this and have specified the ingredients.

Comment 19: 3.4. Lines 127–134: This information is repetitive because it was provided in the Results section.

Response 19: Thank you for the comment. We have rewritten this parts and added more references to support our results.

Comment 20: 3.5. Lines 135–138: Which complex macromolecular compounds can be degraded into micromolecular ones during fermentation?

Response 20: Thank you for the comment. We have rewritten this part and modified our views.

Comment 21: 3.6. Lines 138–150: This is a repetition of the same statements indicated in the previous sentences.

Response 21: Thank you for the comment. We have rewritten this parts and avoided simple duplication of results.

Comment 22: Generally, in this section, the authors continuously repeat the results without clearly explaining the results obtained. In fact, this section is very boring and very difficult to interpret and understand.

In fact, the authors did not explain the increase or decrease of the different active compounds that they studied during fermentations with Bacillus subtilis and Bifidobacterium bifidum.

Response 22: Thank you for the comment. We have rewritten all discussion and tried to discussed in depth. We also added more references to support our results and avoided simple duplication of results.

  1. Materials and Methods

Comment 23: 4.1. Line 155: Why was the resultant flesh mixed with rice wine? Why was the resultant flesh mixed with rice wine at a ratio of 3:1 by mass?

Response 23: Thank you for the comment. These pretreatment processes were carried out according to the traditional treatment methods of COF, as described in the Chinese Pharmacopoeia (2015) and Song et al. (2018). We have cited these documents.

Comment 24: 4.2. Line 170–171: Why was the COF fermented by B. bifidum under anaerobic conditions? Why was the COF fermented by B. subtilis under aerobic conditions?

Response 24: Thank you for the comment. COF were fermented by Bacillus subtilis under aerobic conditions because this bacterium is an aerobic type of bacteria. COF were fermented by Bifidobacterium bifidum under anaerobic conditions because this bacterium is an anaerobic type of bacteria. We have explained this in Materials and Methods.

Comment 25: 4.3. Lines 181–183: Why were different inoculum concentrations of B. bifidum (1 × 107 cfu/mL) and B. subtilis (2 × 107 cfu/mL) used to ferment the COF at different initial concentrations?

Response 25: Thank you for the suggests. First, We had some errors in the print format. “1 × 107 cfu/mL” should be “1 × 107 cfu/mL”, and “2 × 107 cfu/mL” should be “2 × 107 cfu/mL”. Second, the usage of different initial inoculum concentrations of B. bifidum and B. subtilis was due to the fact that the inoculum concentration of B. subtilis and B. bifidum we had prepared previously was 2 × 107 cfu/mL and 1 × 107 cfu/mL respectively.

  1. Conclusions

Comment 26: Lines 251–262: These sentences are repetitive. No clear conclusion has been expressed in this section.

Response 26: Agree. We have rewritten the conclusion and tried to expressed the section.

Sincerely,

Xiuren Zhou

Jan. 5, 2003

Round 2

Reviewer 2 Report

The authors have addressed all the previous comments and the overall quality of the manuscript was improved.

Regarding the newly added parts, the authors should perform a careful proofreading for some typos and italics missing, for example “Helicobacter pylori” in line 32 and “B. subtilis” in line 90; an extra space in line 94; just to name a few.

Figures work quite fine in this new version; however, they should also be mentioned during the discussion, besides Table 1, as they show the main results.

The changes that were made in the material and methods section to use the first person of plural “we” are not necessary and do not improve the manuscript. In many instances the use of the first person should even be avoid. I would suggest that the authors rewrite this as it was in the first version, except if this was made by specific indication of another reviewer or the editor.

Author Response

Dear Reviewer,

We appreciate you for your precious time in reviewing our paper and providing valuable comments. It was your valuable and insightful comments that led to improvements in the current version. The authors have carefully considered the comments and tried our best to address every one of them. We hope the manuscript after careful revisions meet your high standards. The authors welcome further constructive comments if any.

Below we provide the point-by-point responses. All modifications in the manuscript with track changes have been marked up using the“Track Changes”function.

Comment 1: Regarding the newly added parts, the authors should perform a careful proofreading for some typos and italics missing, for example “Helicobacter pylori” in line 32 and “B. subtilis” in line 90; an extra space in line 94; just to name a few.

Response 1: Thank you for pointing this out. We have performed a careful proofreading and corrected the errors found.

Comment 2: Figures work quite fine in this new version; however, they should also be mentioned during the discussion, besides Table 1, as they show the main results.

Response 2: Agree. We have done.

Comment 3: The changes that were made in the material and methods section to use the first person of plural “we” are not necessary and do not improve the manuscript. In many instances the use of the first person should even be avoid. I would suggest that the authors rewrite this as it was in the first version, except if this was made by specific indication of another reviewer or the editor.

Response 3: Agree. We have rewritten this as it was in the first version.

In addition, the English language and style have been carefully checked and revised.

Once again, thank you very much for your comments and suggestions.

Sincerely,

Xiuren Zhou

Jan. 17, 2023

Reviewer 3 Report

After reviewing both the revised paper and the responses provided by the authors, I consider that the manuscript is now acceptable for publication in Molecules. The authors' answers are very clear and respectful and they appropriately addressed the comments of the reviewer to improve the manuscript. They have responded to the different questions raised by the reviewer in a very polite and professional way. Congratulations!!

Author Response

Dear Reviewer,

We appreciate you for your precious time in reviewing our paper and providing valuable comments. It was your valuable and insightful comments that led to improvements in the current version. The authors have carefully considered the comments and tried our best to address every one of them. We hope the manuscript after careful revisions meet your high standards. The authors welcome further constructive comments if any.

Below we provide the point-by-point responses. All modifications in the manuscript with track changes have been marked up using the“Track Changes”function.

Comment 1: After reviewing both the revised paper and the responses provided by the authors, I consider that the manuscript is now acceptable for publication in Molecules. The authors' answers are very clear and respectful and they appropriately addressed the comments of the reviewer to improve the manuscript. They have responded to the different questions raised by the reviewer in a very polite and professional way. Congratulations!!

Response 1: We are very grateful to you for your time and comments.

In addition, the English language and style have been carefully checked and revised.

Once again, thank you very much for your comments and suggestions.

Sincerely,

Xiuren Zhou

Jan. 17, 2023